# Jagged and Delta-like ligands control distinct events during airway progenitor cell differentiation

**Maria R Stupnikov[1,2], Ying Yang[1,2], Munemasa Mori[1,3], Jining Lu[1,3], Wellington V Cardoso[1,2,3]\***

[1]Columbia Center for Human Development, Department of Medicine, Columbia University Medical Center, New York, United States; [2]Department of Genetics and Development, Columbia University Medical Center, New York, United States; [3]Division of Pulmonary Allergy and Critical Care Medicine, Department of Medicine, Columbia University Medical Center, New York, United States

**Abstract** Notch signaling regulates cell fate selection during development in multiple organs including the lung. Previous studies on the role of Notch in the lung focused mostly on Notch pathway core components or receptor-specific functions. It is unclear, however, how Jagged or Delta-like ligands collectively or individually (Jag1, Jag2, Dll1, Dll4) influence differentiation of airway epithelial progenitors. Using mouse genetic models we show major differences in Jag and Dll in regulation and establishment of cell fate. Jag ligands had a major impact in balancing distinct cell populations in conducting airways, but had no role in the establishment of domains and cellular abundance in the neuroendocrine (NE) microenvironment. Surprisingly, Dll ligands were crucial in restricting cell fate and size of NE bodies and showed an overlapping role with Jag in differentiation of NE-associated secretory (club) cells. These mechanisms may potentially play a role in human conditions that result in aberrant NE differentiation, including NE hyperplasias and cancer.

**\*For correspondence:**
wvc2104@cumc.columbia.edu

## Introduction

Notch signaling is a major regulator of progenitor cell fate and differentiation during organogenesis, repair-regeneration, and cancer. In mammals, four Notch receptors (Notch1–4) and five ligands (Delta-like: Dll1, Dll3 and Dll4 and Jagged: Jag1 and Jag2) have been described. All ligands, except Dll3, are Notch activating. Signaling is triggered by ligand-receptor binding through cell-cell interactions, which leads to sequential cleavage of the Notch receptor and binding of its intracellular domain (NICD) to a CSL/RBPJk-activator complex for activation of downstream target genes, such as HEY/HES-family members (*Radtke and Raj, 2003*; *Bray, 2006*). While different Notch receptors are known to act in a variety of biological processes, evidence from genetic studies suggest that the Notch effects are not necessarily dependent on the type of NICD but rather of NICD dosage (*Liu et al., 2015*). Notably, specific Notch ligand-receptor binding in mammalian cells appears to be mostly non-selective or context-dependent.

Interestingly, systemic deletion of *Jag1*, *Jag2*, or *Dll4* has been shown to result in distinct phenotypes, suggesting that these ligands could mediate unique functions not entirely due to the receptor they activate (*D'Souza et al., 2009*; *Choi et al., 2009*). Indeed, Notch ligands were reported to activate distinct targets even through binding to the same Notch receptor and ligand-specific effects have been observed in multiple contexts (*Nandagopal et al., 2018*).

The Notch pathway plays a crucial role in the developing lung. When airways are still forming epithelial progenitors initiate a differentiation program that gives rise to secretory (club, goblet),

**eLife digest** Cells communicate with each other by sending messages through a range of signaling pathways. One of the ways cells signal to each other is through a well-studied pathway known as Notch. In this pathway, cells display molecules on their surface, known as Notch ligands, that can activate Notch receptor proteins on the surface of neighboring cells. Once the Notch receptors bind to these ligands, they trigger various responses inside the cell. Notch ligands exist in two different families: Delta-like (Dll) ligands and Jagged (Jag) ligands.

The layer of cells that lines the airways in the lungs consists of several different cell types. These include secretory cells that produce the fluid covering the airway surface, multiciliated cells, and neuroendocrine cells. Together these cells work as a barrier to protect the lung from environmental particles that may be breathed in. Additionally, the lung also has multipotent progenitor cells, which can become any of the other types.

When Notch signaling is missing from the lung during embryonic development, not enough secretory cells are made, while other cell types are made in excess. This is because the multipotent progenitor cells need to communicate via Notch signaling to decide what type of cell to become and keep the right proportion of different cell types in the airways. In other organs, multipotent progenitors can become different types of cells depending on whether Notch signaling was activated by Dll or by Jag ligands, but it was unknown if this also happened in the lungs.

Stupnikov et al. investigated the situation in the airways during development by looking at where and when Dll and Jag ligands first appeared, and by inactivating the genes that code for these ligands. They found that Jag ligands appeared well before Dll ligands, and that when the genes coding for Jag ligands were inactivated, more ciliated cells were produced. By contrast, loss of Dll ligands resulted in an increase in the neuroendocrine and their associated secretory cells, with little effect on the multiciliated cells. This increase resembled what is seen in some human diseases.

The results suggest that the diversity of Notch effects in the airways depends on which Notch ligand is locally available. These observations may help to understand the mechanism of certain diseases involving neuroendocrine cells in the lung, such as small cell carcinoma or bronchial carcinoid tumors.

multiciliated, and neuroendocrine (NE) cells. Previous studies addressing the role of Notch in the lung focused largely on central components of this pathway (Rbpjk, Pofut1, and Hes1). Disruption of Rbpjk or the o-fucosyl-transferase Pofut1 required for Notch signaling results in aberrant expansion of multiciliated and NE cells at the cost of secretory cells (*Tsao et al., 2009*; *Tsao et al., 2011*; *Morimoto et al., 2010*). Subsequent studies showed that club cells are more sensitive to deficiency in Notch2 while Notch 1-3 receptors contribute to control the NE population in an additive manner (*Morimoto et al., 2012*). However, it was unclear whether individual ligand families (Delta-like and Jagged) or specific ligands (Dll1, Dll4, Jag1, and Jag2) influence distinct aspects of differentiation of airway epithelial progenitors. Notably, these ligands have been reported in partially overlapping but also distinct domains in the lung (*Post et al., 2000*; *Kong et al., 2004*; *Tsao et al., 2009*; *Xu et al., 2010b*; *Zhang et al., 2013*; *Mori et al., 2015*).

Here we explored the role of ligands using single and double conditional Jagged and Delta-like null alleles targeted to epithelial progenitors from early lung development. We show remarkably distinct roles of these ligands in the developing intra- and extrapulmonary airways and in the control of the expansion and differentiation of the NE microenvironment.

## Results

### Jagged ligands precede the appearance of Delta-like ligands in differentiating airway progenitors

Although the expression patterns of Jag and Dll have been reported in both epithelial and mesenchymal layers of the developing lung, specific information about their onset of expression and regional distribution in the epithelial compartment at early stages of differentiation has been scattered and not well integrated to functional studies (*Post et al., 2000*; *Kong et al., 2004*; *Xu et al.,*

*2010b*; *Morimoto et al., 2012*; *Tsao et al., 2009*). To gain further insights into this issue we revisited the spatial and temporal pattern of expression of Notch ligands when epithelial cells are initiating commitment to different cell fates in developing airways.

By in situ hybridization (ISH) analysis none of these ligands were detectable in the airway epithelium prior to or at E11.5 (not shown). However, at around E12.0 evidence of *Jag2* epithelial signals in the developing trachea made it the first of all Notch ligands to be induced in the differentiation program of airways (*Figure 1A*). Expression progressed in a proximal-to-distal fashion; at E12.5 low level signals were detected in the epithelium of extrapulmonary but not intrapulmonary airways. This contrasted with the strong *Jag2* signals present in the esophageal epithelium and in neighboring vascular structures (*Figure 1B*). Notably, the *Jag2* detection in epithelial progenitors of the trachea and extrapulmonary airways coincided with the previously reported onset of Notch activation and appearance of the secretory cell marker *Scgb3a2* locally (*Guha et al., 2012*). No epithelial *Jag1* could be detected anywhere in airways at these stages, although clearly present in vascular structures (*Figure 1C*). These data supported the idea of a Jag2-Notch program giving rise to secretory cell precursors as one of the earliest events initiating differentiation in airways, even preceding the appearance of pulmonary NE cells (PNEC) reported to begin only within a day later (*Li and Linnoila, 2012*; *Kuo and Krasnow, 2015*; *Noguchi et al., 2015*; *Sui et al., 2018*). Indeed, expression of *Ascl1*, which marks PNEC precursors, was first found ~E13-13.5 in large intrapulmonary airways and both *Dll1* and *Dll4* were then subsequently expressed in these precursors (*Figure 1E,D*). By E13.5-E14.5 strong *Jag2* epithelial signals were seen throughout the trachea and main bronchi, in contrast to *Jag1*, nearly undetected at these sites (*Figure 1*, *Figure 1—figure supplement 1*). At E14.5 NEBs and PNECs were sharply demarcated by Ascl1, and *Dll1* and *Dll4* transcripts became prominently expressed in NEBs (*Figure 1E–F*). This was accompanied by the appearance of clusters of cells adjacent to NEBs, collectively marked by expression of the uroplakin *Upk3a*, the cell surface stem cell (secretory) marker SSEA1, the secretoglobin Scgb3a2 and low levels of the cytochrome gene *Cyp2f2* and CC10. The pattern was consistent with the initiation of a Notch-dependent program of secretory cells in the NEB microenvironment (*Guha et al., 2012* and described later).

Thus, Jag and Dll ligands appear in different domains and in a sequential proximal-distal fashion during the establishment of cell fate in airway progenitors, initiating with *Jag2* in the trachea, *Jag1*, and lastly *Dll1* and *Dll4* once NEBs form in intrapulmonary airways (summary diagram *Figure 1G*).

## Jag1 and Jag2 regulate the balance of different cell types in extra- and intrapulmonary airways

Given the distinct timing and spatial distribution of Jag ligands described above, we reasoned that common but also non-overlapping functions were likely to exist in the distinct domains of the respiratory tract. Inactivation of Jag1 in epithelial progenitors of intrapulmonary airways undergoing branching morphogenesis using a surfactant protein-c (*Sftpc*)-tet-O system was shown to disrupt epithelial differentiation (*Zhang et al., 2013*), confirming the previously reported role of Notch signaling in this process (*Tsao et al., 2009*; *Morimoto et al., 2010*). Although efficient, in this targeting strategy Cre-mediated recombination was restricted to intrapulmonary airways, initiating at the onset of *Sftpc* expression in secondary buds (~E10.5). Thus, information about a putative role of Jag1 in extrapulmonary airways (trachea, main bronchi) and at stages prior to E10.5 was missing. Moreover, little was known about how Jag2 influences lung development and whether there is any functional overlap between Jag2 and Jag1. Jag2 systemic knockout animals die at birth (*Jiang et al., 1998*). Lastly, no information was available about compensation of Jag by other ligands during epithelial differentiation.

We used the Shh-cre line to inactivate Jag1 and Jag2 individually or in combination in early epithelial progenitors of both extrapulmonary and intrapulmonary at the onset of lung development (*Harfe et al., 2004*). Jag1$^{flox/flox}$; Shh$^{cre/+}$ (Jag1$^{cnull}$), Jag2$^{flox/flox}$; Shh$^{cre/+}$ (Jag2$^{cnull}$) and double (Jag1$^{cnull}$; Jag2$^{cnull}$) null mutants were analyzed at early (E14.5) and late (E18.5) stages of airway differentiation. Gross morphological analysis of the mutant lungs showed no notable macroscopic difference in size or shape (not shown). We compared the effects of Jag1 and Jag2 loss in multiciliated-secretory cell fate selection at E18.5, once differentiated cell profiles were largely established in extrapulmonary (trachea) and intrapulmonary (lung) airways. qPCR analysis of E18.5 lung homogenates showed significant changes in markers of epithelial differentiation in all mutants (*Figure 2A*). Expression of the secretory markers Scgb3a2 and Scgb1a1 (encoding CC10) were reduced by 86.2%

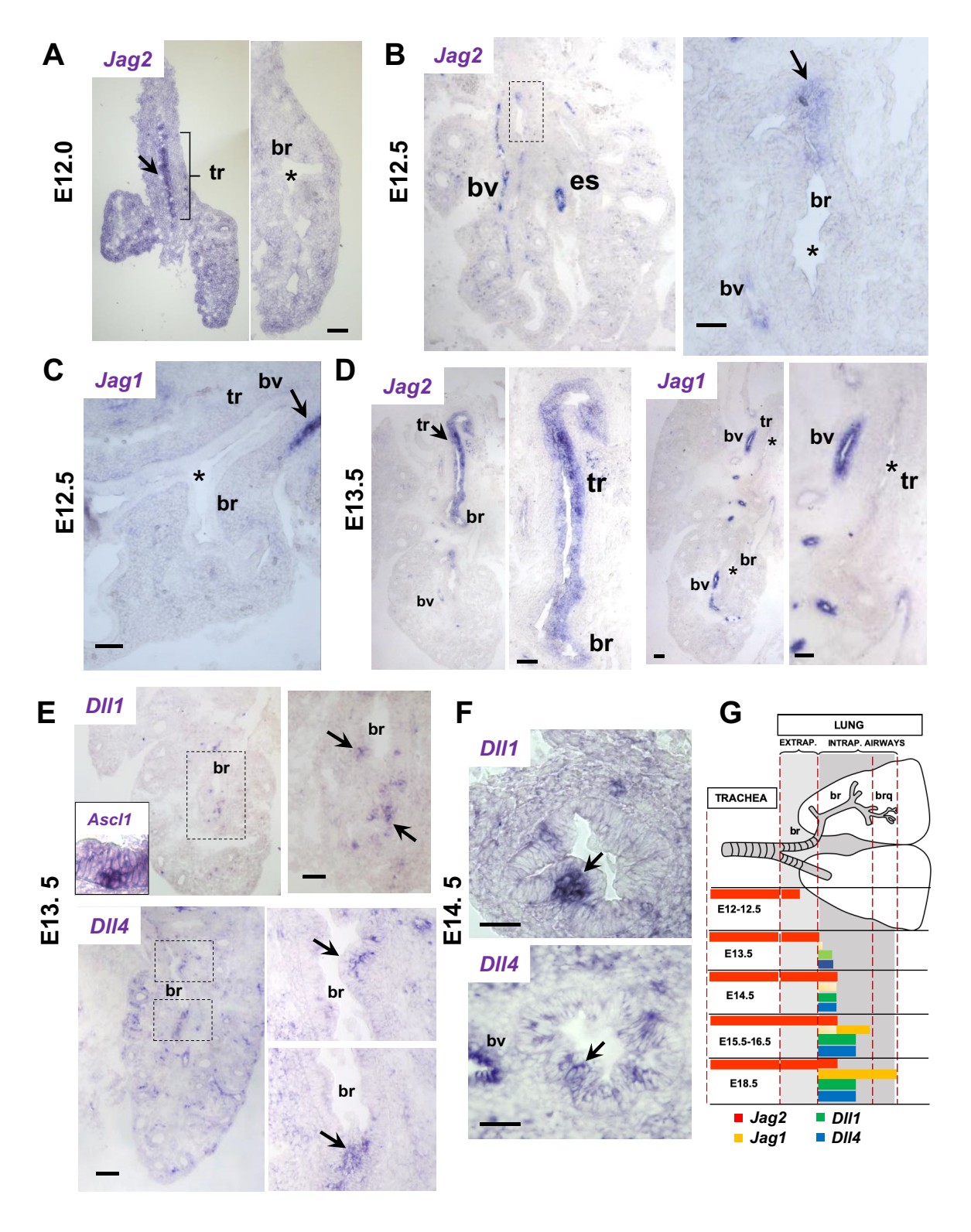

**Figure 1.** Jag and Dll ligands arise in distinct spatial and temporal patterns in airway epithelial progenitors undergoing cell fate commitment. In situ hybridization of E12-E14.5 lungs. (A–C) E12-E12.5: *Jag2* is the first and only Notch ligand detected in the epithelium, initially restricted to the trachea (tr) and later extending to extrapulmonary airways but not intrapulmonary large airways (bronchi: br)(B, boxed area enlarged in right panel). *Jag1 is* absent from both trachea and bronchi. Strong Jag ligand signals in blood vessels (bv) and the esophageal epithelium (es). (D) E13.5: Prominent epithelial *Jag2*

*Figure 1 continued on next page*

*Figure 1 continued*

signals still restricted to trachea and extrapulmonary bronchi; abundant *Jag1* in vascular but not in the epithelial compartment. (**E–F**) E13.5- E14.5: *Dll1* and *Dll4* signals are first detected at E13.5 in epithelial cell clusters of intrapulmonary main bronchi at sites of NEB formation marked by *Ascl1* (inset) and signals become highly localized and prominent from E14.5 onward (**G**) Schematic summarizing the time and sites of Notch ligand expression throughout the developing respiratory tract epithelium. Arrows depict representative signals in airway epithelium; (*) depicts absence or near background signal. Bars in A-F = 40 μm.

The online version of this article includes the following figure supplement(s) for figure 1:

**Figure supplement 1.** Distinct spatial and temporal patterns of *Jag2* and *Jag1* in the developing airway epithelium.

**Figure supplement 2.** NEBs emerge in non-Jag expressing domains of the developing airway epithelium.

(p=$5\times10^{-12}$) and 85.6% (p=0.0001), respectively in Jag1$^{cnull}$ mutants, but only by 25.9% (p=0.015) and 34.4% (p=0.019), respectively in Jag2$^{cnull}$ mutants. These changes were accompanied by a significant increase in Foxj1 expression in Jag1$^{cnull}$ (183% increase, p=0.0006) but not in Jag2$^{cnull}$ (13% reduction, p=0.501) mutants. Thus, the differentiation program of intrapulmonary airways was more severely affected in Jag1$^{cnull}$ than in Jag2$^{cnull}$ mutants. The predominant contribution of Jag1 to the program of secretory cell fate as represented by these markers could be clearly seen in double Jag1$^{cnull}$; Jag2$^{cnull}$ mice. These mutants showed Scgb3a2 and Scgb1a1 nearly abolished and an increase in Foxj1 similar to that found in Jag1$^{cnull}$. Altogether these results indicated that airway progenitors are largely dependent on Jag ligands to initiate secretory cell differentiation.

Immunofluorescence of Foxj1 and CC10 in E18.5 lung sections confirmed the changes in gene expression in intrapulmonary airways revealed by qPCR and showed secretory cells less abundant in Jag1$^{cnull}$ compared to Jag2$^{cnull}$ mutants (*Figure 2B*). Interestingly, multiciliated cell fate appeared to be minimally affected in Jag2$^{cnull}$ airways. Morphometric analysis showed no significant change in the number of Foxj1+ cells in Jag2$^{cnull}$ airways relative to control (p=0.164), in contrast to the ~2.5 fold increase in these cells in Jag1$^{cnull}$ mutants (p=$4.17\times10^{-7}$) (*Figure 2C*).

To search for potential reasons contributing to the more severe ciliated cell phenotype in Jag1$^{cnull}$ compared to Jag2$^{cnull}$ mutants we further extended our analysis of Notch ligands to later developmental stages (*Figure 1—figure supplement 1*). Interestingly, ISH of E14.5- E18.5 WT lungs showed that *Jag2* expression in the trachea and extrapulmonary airways continued to be robust at later stages while remaining weak and scattered in intrapulmonary bronchial epithelia. By contrast, *Jag1* expression was progressively stronger in the intrapulmonary airway epithelium from E15.5 onwards and by E18.5 expression extended to the distal bronchioles. Double ISH-immunohistochemistry for Foxj1 confirmed our previous report of *Jag1* localization in multiciliated cells (*Tsao et al., 2009*). Thus, our data suggested that Jag1 is the predominant ligand in intrapulmonary airways mediating secretory vs. multiciliated cell fate choice during differentiation. Consequently, Jag1$^{cnull}$ mutants were expected to display the unbalanced abundance of multiciliated cells in intrapulmonary airways compared to Jag2$^{cnull}$ mutants. We also reasoned that in Jag2$^{cnull}$ mutants Notch-mediated signaling by Jag1 should compensate for the loss of Jag2, preserving the balance of multiciliated vs. secretory cell differentiation in these mutants. Indeed, analysis of Jag2$^{cnull}$ mice confirmed the presence of *Jag1* signals in the intrapulmonary airway epithelium of these mutants (*Figure 2D*).

IF staining of Krt5 and β-tubulin in E18.5 Jag1$^{cnull}$; Jag2$^{cnull}$ double mutants revealed an expansion in the population of basal progenitors and a variable but also increased population of multiciliated cells (*Figure 2—figure supplement 1A*). Single Jag1$^{cnull}$ or Jag2$^{cnull}$ mutants were then examined to assess the contribution of each Jag ligand to the double mutant phenotype. For morphometric analysis we first performed IF for p63 and Foxj1, which label basal and multiciliated cells, respectively, and have the advantage of displaying nuclear signals, thus facilitating quantitation. Tracheal sections from all groups at E18.5 were analyzed and the %p63+/DAPI and %Foxj1+/DAPI were estimated. This analysis revealed that Jag2$^{cnull}$ tracheas had a significant increase in the number of prebasal cells (%p63+/DAPI) but a trend (not significant) towards an increase in the abundance of the multiciliated cell population (*Figure 2—figure supplement 1B,C*). Notably, double ISH/immunohistochemistry for *Jag2*/p63 in WT confirmed the presence of *Jag2* in prebasal cells of the developing trachea (*Figure 1—figure supplement 1*; *Mori et al., 2015*). As expected from the distinct spatial distribution of these Jag ligands, single Jag1$^{cnull}$ tracheas had no detectable change in the number of basal cell progenitors compared to controls.

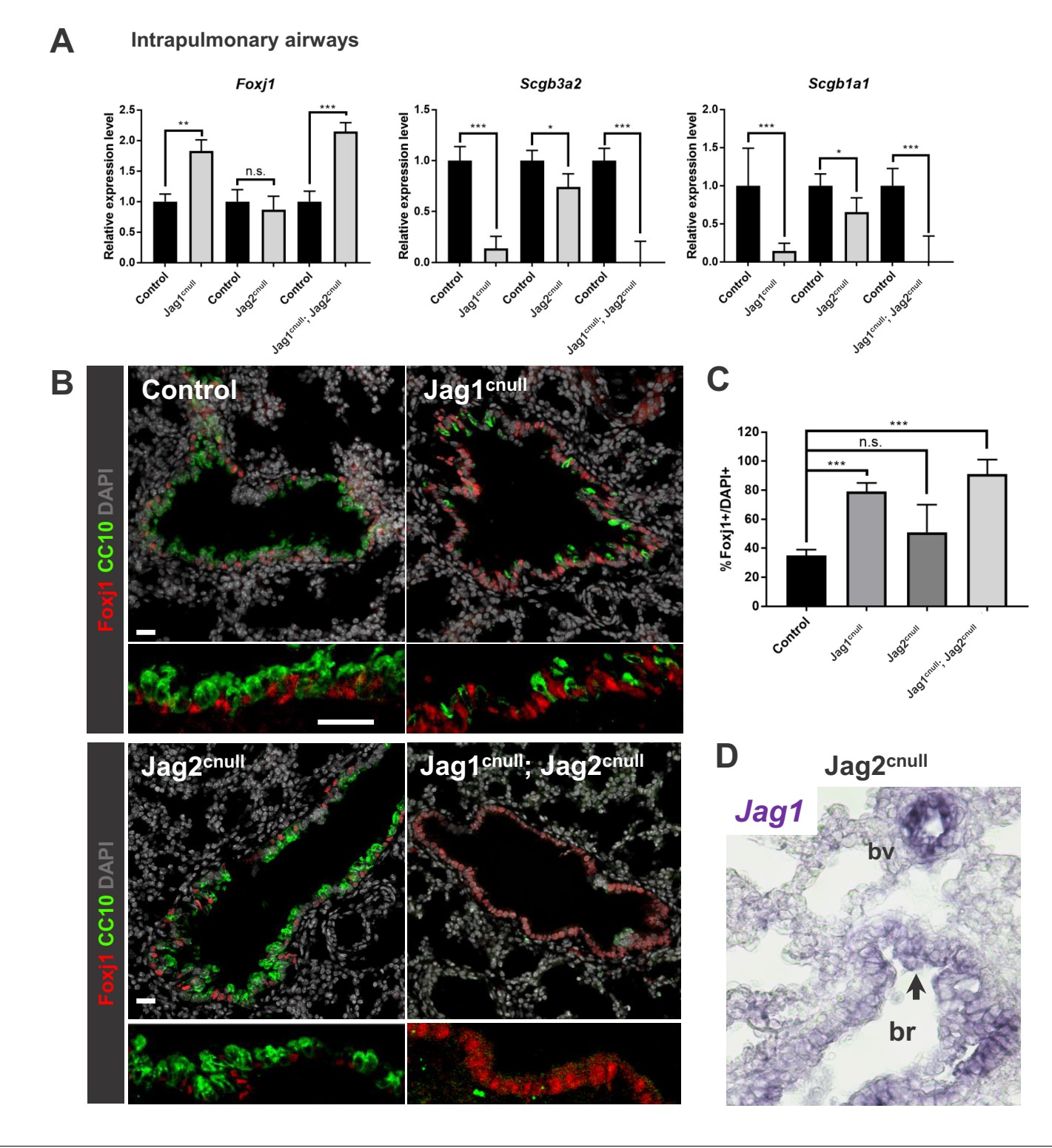

**Figure 2.** *Jag1* and *Jag2* collectively contribute to balance multiciliated and secretory cell fate in intrapulmonary airways. (**A**) qPCR analysis: markers of multiciliated (*Foxj1*) and secretory cell fate (*Scgb3a2, Scgb1a1* [*CC10*]) in mutant and respective control lung homogenates (n = 4 *Jag1* control, n = 3 Jag1cnull; n = 4 *Jag2* control; n = 4 Jag2cnull; n = 4 *Jag1/Jag2* control; n = 4 Jag1cnull; Jag2cnull). Decreased expression of secretory markers and increased *Foxj1* predominantly in Jag1cnull and in double Jag1cnull; Jag2cnull. Graphs represent mean ± SEM. Student's t-test: differences statistically significant at *p<0.05, **p<0.005, ***p<0.005, n.s., not significant. (**B**) Immunofluorescence of lungs from E18.5 control and Jag mutant mice showing changes in expression of Foxj1 and CC10 in intrapulmonary airways consistent with that revealed by qPCR (DAPI in gray). (**C**) Morphometric analysis:
*Figure 2 continued on next page*

*Figure 2 continued*

percentage of Foxj1+ cells in control and mutant intrapulmonary airways (normalized by DAPI). Significant increase in the number of Foxj1 labeled cells in Jag1$^{cnull}$ and in Jag1$^{cnull}$; Jag2$^{cnull}$, but not in Jag2$^{cnull}$ mutants. (D) In situ hybridization of *Jag1* in E18.5 Jag2$^{cnull}$ showing epithelial signals in intrapulmonary airways (arrow, bronchi: br) and in blood vessels (bv). Scale bar in B = 40 μm.

The online version of this article includes the following figure supplement(s) for figure 2:

**Figure supplement 1.** Loss of Jag ligands results in expansion of the embryonic basal cells and multiciliated cells in the E18.5 extrapulmonary airways (trachea).

Together these data suggest that Jag ligands mediate overlapping but distinct events along the respiratory tract epithelium. In extrapulmonary airways (trachea here) Jag2 predominantly contributes to the balance of basal versus luminal cells while Jag1 controls abundance of multiciliated cells. By contrast, in intrapulmonary airways Jag1 is the predominant ligand regulating the balance of multiciliated versus club cell fate with a lesser contribution of Jag2.

## Jag ligands have minimal effects in the establishment and regulation of the NE microenvironment

Next we investigated whether Jag ligands could influence cell fate events that ultimately regulate the PNEC pool in intrapulmonary airways, regardless of its organization as NEBs or as solitary cells. Thus, we compared levels of *Ascl1* expression in homogenates of *Jag*-cnull mutant lungs at E18.5, when NEBs are already widely distributed at branch point and internodal locations. qPCR analysis showed no difference in *Ascl1* expression between controls and mutants in any of the *Jag*-deficient airways (*Figure 3A*). Consistent with this, immunofluorescence for Cgrp, another established marker of PNEC fate (*Li and Linnoila, 2012*), did not reveal consistent differences in expression patterns, suggestive of alterations in NEBs' spatial distribution, size (~8–10 PNECs per NEB control, Jag1$^{cnull}$, Jag2$^{cnull}$, and double Jag1$^{cnull}$; Jag2$^{cnull}$) or frequency in intrapulmonary airways of mutants compared to controls (*Figure 3A–D*). Since in our mutants *Jag* is deleted well before epithelial progenitors differentiate, we concluded that Jag-mediated Notch signaling is unlikely to be involved in the mechanisms that initiate or restrict the domains of NEB or PNEC fate. Neither NEBs nor PNECs could be identified in extrapulmonary airways (trachea) of mutants or control animals.

We then examined the effect of *Jag* deletion in the population of secretory progenitors tightly associated with the developing NEBs. Previous studies have shown that they arise around E14.5 immediately adjacent to Ascl1-expressing cell clusters, being distinguished from other secretory precursors collectively by their expression of *Upk3a*, SSEA1, Scgb3a2, and low levels of *Cyp2f2* and CC10 (*Guha et al., 2012*; *Morimoto et al., 2012*). Immunofluorescence of E18.5 lung sections triple-labeled with SSEA1, Scgb3a2, and Cgrp identified the typical SSEA1+ Scgb3a2+ cells around Cgrp+ clusters similarly preserved in intrapulmonary airways of Jag1$^{cnull}$ and Jag2$^{cnull}$ mutants (*Figure 3B–D*). This contrasted with the nearly absent expression of these markers outside the NEB microenvironment in Jag1$^{cnull}$; Jag2$^{cnull}$ mutants (*Figure 3B–C*, described above). Remarkably, in double Jag1$^{cnull}$; Jag2$^{cnull}$ airways the only population of cells expressing secretory markers was that associated with NEBs (*Figure 3C,D*). This NEB-associated cell population was heterogeneous in regards to expression of the markers above even in the same airway, as also observed in control lungs (*Figure 3C*). Notably, double Jag1$^{cnull}$; Jag2$^{cnull}$ showed no evidence of change in the size of this population relative to NEBs. Since these cells are crucially dependent on Notch signaling and their only source of ligand should be Dll1 and/or Dll4 from PNECs, we performed N1ICD immunofluorescence to examine the status of Notch activation locally. Strong N1ICD labeling was found selectively in the SSEA1+ NEB-associated cell populations of mutants, indistinguishable from controls (*Figure 3D*). Lastly, qPCR analysis of *Upk3a*, a gene marker highly enriched in NEB-associated secretory cells but that also labels scattered secretory cells of intrapulmonary airways, showed markedly decreased levels of expression in double Jag1$^{cnull}$; Jag2$^{cnull}$ lungs (*Figure 3E*). This was consistent with the dependence of *Upk3a* on Notch signaling we previously reported (*Guha et al., 2012*). The similar levels of *Ascl1* and the Cgrp expression pattern we found in controls and double *Jag* mutants (*Figure 3A*) suggested that the NEB size and frequency in airways is not dependent on Jag ligands. Thus we reasoned that the remaining population of *Upk3a*-expressing cells in Jag1$^{cnull}$; Jag2$^{cnull}$ mutants was associated with NEBs and that the significant decrease in *Upk3a* seen by qPCR resulted

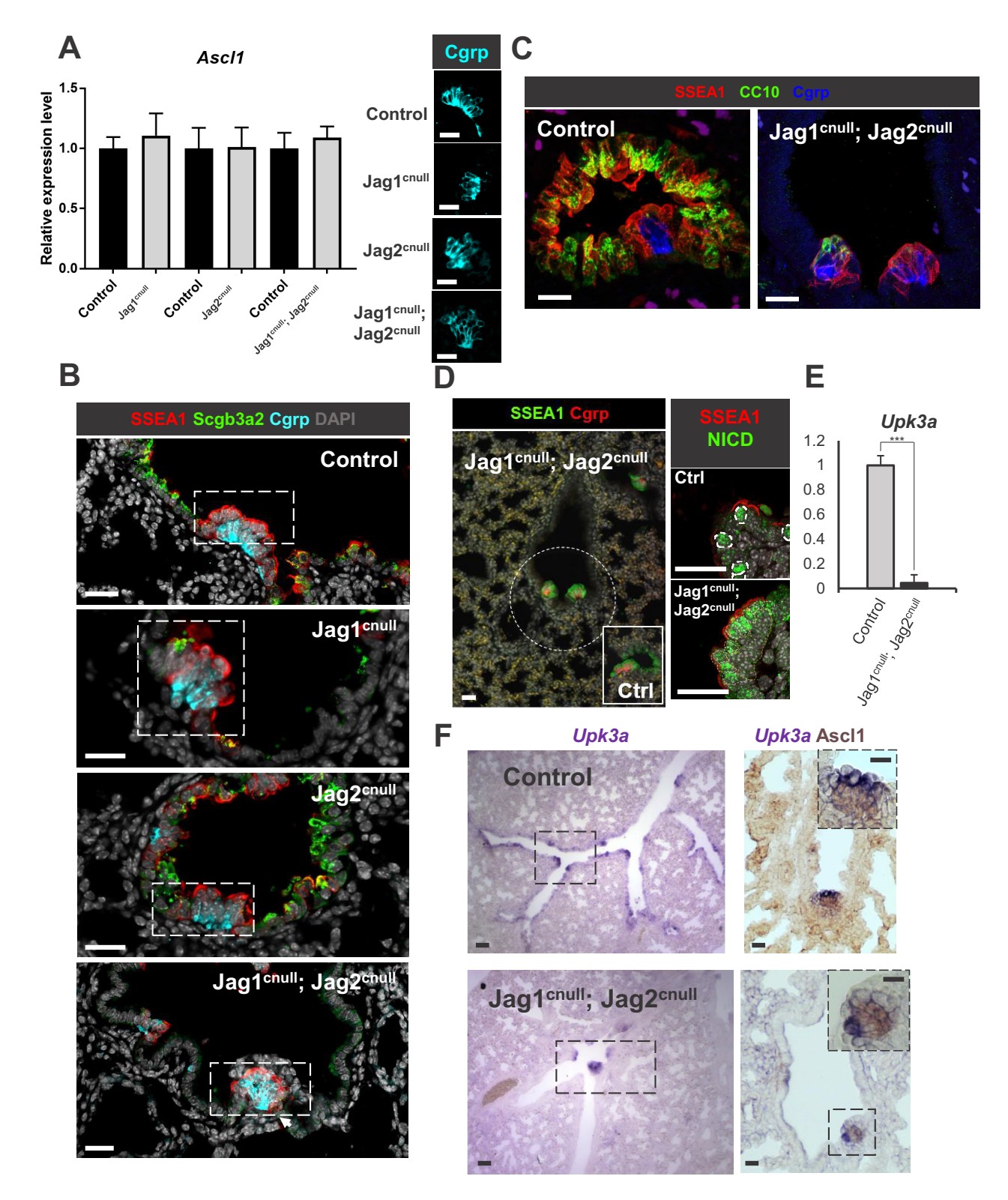

**Figure 3.** Disruption of Jagged-driven Notch signaling has no detectable impact on PNEC and NEB cell fate specification or maintenance. (**A**) qPCR analysis of *Ascl1* in E18.5 controls, single and double Jag1[cnull]; Jag2[cnull] mutants showing no significant difference in expression. Graph represents mean ± SEM (n = 4 *Jag1* control, n = 3 Jag1[cnull]; n = 4 *Jag2* control; n = 4 Jag2[cnull]; n = 4 *Jag1/Jag2* control; n = 4 Jag1[cnull]; Jag2[cnull]). Student's t-test was used to analyze data. Side panels: representative Cgrp immunofluorescence (IF) in controls and mutants (**B**) IF of secretory (Scgb3a2), NEB-

*Figure 3 continued on next page*

*Figure 3 continued*

associated SSEA1 (CC) and Cgrp (PNEC/NEB) in control and Jag mutants. (**C, D**) Preserved NEB microenvironment in E18.5 Jag1$^{cnull}$; Jag2$^{cnull}$ double mutants: IF showing NEBs (Cgrp) and NEB-associated CCs (SSEA1, N1ICD, CC10$^{low}$) in double null mutants similar to controls (Ex. inset in D). (**E**) qPCR analysis of *Upk3a* in E18.5 control and double Jag1$^{cnull}$; Jag2$^{cnull}$ mutants showing nearly abolished *Upk3a* expression in mutants (n = 3 in each group). Graph: mean ± SEM; ***p<0.0005 (**F**) ISH of *Upk3a* (left panels) and double labeled with Ascl1 (immunohistochemistry, right panels): *Upk3a* expression predominantly in NEB-associated CCs with rare signals scattered in CC elsewhere. In double Jag1$^{cnull}$; Jag2$^{cnull}$ mutants CCs are abolished but in the NEB microenvironment and *Upk3a* becomes restricted to those surrounding NEBs. Scale bar in A-D, F = 40 µm. Scale bar in F inset = 20 µm.

from the loss of the *Upk3a*-expressing scattered cell population. Indeed, ISH of *Upk3a* in these mutants showed signals restricted to branch points in proximal regions of intrapulmonary airways associated with Ascl1-expressing NEBs (*Figure 3F*). This was further supported by the evidence of a preserved NEB-associated SSEA1 population (*Figure 3B–D*) and the fact that *Upk3a* requires Notch activation not present in the extensive areas devoided of Jag ligands.

Together these results suggested that Jag1 and Jag2 have overlapping but also distinct roles in the cell fate specification of respiratory lineages in extrapulmonary and intrapulmonary airways. Jag ligands, however, appear to be dispensable for activation of Notch and induction of NEB-associated secretory cells since they can utilize Dll provided by their neighboring NE cells. Moreover, our data show no evidence that Jag ligands have any impact in regulating size or frequency of NEBs.

## Dll ligands control the size of the NEB microenvironment

Our analysis of Jag1$^{cnull}$; Jag2$^{cnull}$ mutants identified seemingly self-contained units comprised of Dll-expressing NEBs and immediately adjacent cells able to activate and maintain robust Jag-independent Notch signaling for local secretory differentiation. Previous studies in *Ascl1*$^{-/-}$ mice showed that these units were strictly dependent on the presence of NEBs (*Guha et al., 2012*). Questions remained whether preventing NEBs from expressing Dll ligands would have any impact on the NEB microenvironment or elsewhere if Jag ligands were still expressed. Unlike *Jag1* and *Jag2*, found in largely non-overlapping spatial and temporal patterns, *Dll1* and *Dll4* are collectively expressed in a very restricted fashion to PNEC/NEBs. Given the high probability of functional overlap, we generated mouse mutants in which both Dll ligands were deleted conditionally in the developing lung epithelium. Double deletion (Dll1$^{cnull}$; Dll4$^{cnull}$) was achieved from early stages using a similar targeting strategy with a *Shh*$^{cre/+}$ line.

IF analysis of E14.5 lungs from control mice showed the solitary PNECs and distinct small clusters of Ascl1+ cells in the epithelium of large intrapulmonary airways (bronchi) characteristic of the NEBs. By contrast, Dll1$^{cnull}$; Dll4$^{cnull}$ E14.5 showed a striking expansion in the population of Ascl1+ cells (*Figure 4A*). Although individual Ascl1+ cell clusters could still be identified, they often seem to coalesce in large patches to form a nearly continuous layer of Ascl1+ cells. In spite of the distribution in wider domains, these cells were not found ectopically in extrapulmonary airways or in the most distal airways undergoing branching morphogenesis. The large patches of Ascl1+ cells were identified at branch points in E18.5 lungs and by then co-expressed Cgrp, indicating their continued differentiation (*Figure 4B*). Thus, loss of Dll ligands expanded the Ascl1+ pool of PNEC/NEB precursors and did not prevent these cells from initiating maturation. Interestingly, Ki67 staining showed no difference in labeling associated with the Ascl1-expressing cells between control and double mutants at E18.5 or E14.5 (*Figure 4C*). The data supported the idea that the NEB expansion found in Dll1$^{cnull}$; Dll4$^{cnull}$ mutants did not result from an increase in proliferation.

To assess the contribution of each of these ligands to the Dll1$^{cnull}$; Dll4$^{cnull}$ phenotype, we examined Dll1$^{cnull}$ and Dll4$^{cnull}$ individual mutants. E18.5 lungs were isolated from single and double mutants and changes in expression of *Ascl1* and *Cgrp* were analyzed by qPCR in homogenates (*Figure 4D*). Double *Dll* mutants showed a significant increase in these transcripts compared to controls (*Ascl1 p*=1.1×10$^{-6}$; *Cgrp p*=5.9×10$^{-5}$), consistent with the aberrant NE cell expansion. However, in single Dll1$^{cnull}$ or Dll4$^{cnull}$ lungs *Ascl1 expression* was modestly increased only in Dll1$^{cnull}$ and *Cgrp* mRNA was not altered in either of these mutants compared to controls. The marked difference in phenotype between double and single *Dll* mutants suggested functional redundancy between Dll1 and Dll4 in controlling NEB or PNEC-associated events. To better understand these events, we performed morphometric analysis of the NE compartment in E14.5 lungs to determine the impact of Dll in the size and frequency of NEBs and PNECs (*Figure 4E*). Quantitation of the number of solitary

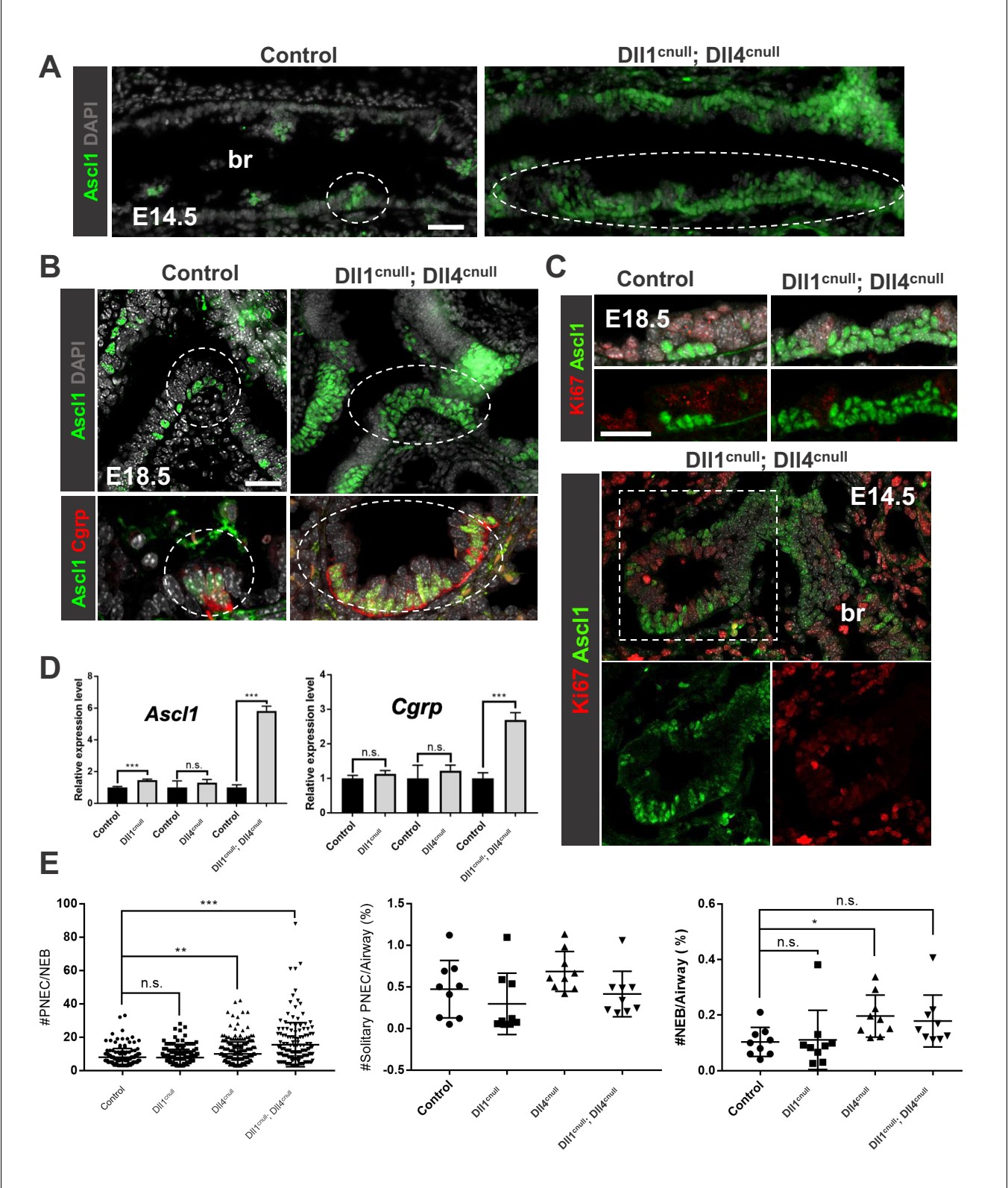

**Figure 4.** Loss of Dll-driven Notch signaling results in expansion of the NEBs. (**A–B**) Ascl1 immunofluorescence (IF) in E14.5 control lungs (**A**) showing discrete clusters of Ascl1+ NEBs in large intrapulmonary airways (bronchi: br) in contrast to the aberrant NEB expansion in Dll1cnull; Dll4cnull lungs. At E18.5 (**B**) NEBs at branchpoints are also enlarged in mutants compared to controls and express Cgrp (bottom panel). (**C**) Double Ki67; Ascl1 IF shows no evidence that NEB expansion results from increased proliferation at E18.5 (top) or E14.5 (bottom panel). (**D**) qPCR analysis of NE markers at E18.5:
*Figure 4 continued on next page*

*Figure 4 continued*

significant increase in *Ascl1* and *Cgrp* expression in double null mutants compared to controls; single mutant Dll1$^{cnull}$ but not Dll4$^{cnull}$ showed increased expression of *Ascl1* (n = 3 *Dll1* control, n = 5 Dll1$^{cnull}$ for *Ascl1* and n = 4 *Dll1* control, n = 5 Dll1$^{cnull}$ for *Cgrp*; n = 3 *Dll4* control n = 6 Dll4$^{cnull}$ for *Ascl1* and *Cgrp*; n = 3 *Dll1/Dll4* control; n = 3 Dll1$^{cnull}$; Dll4$^{cnull}$ for *Ascl1* and *Cgrp*). Graphs: mean ± SEM. ***p<0.0005; n.s., not significant by Student's t-test (E) Morphometric analysis of PNECs and NEBs in E14.5 control and Dll mutant lungs. Left panel: NEB size as determined by number of PNECs per NEB. Center panel: solitary PNEC per airway. Right panel: number of NEBs. * *P<0.05,* ** *P<0.005,* *** *P<0.0005;* n.s., not significant by Student's t-test. Scale bars in A, B = 40 μm.

PNECs in the airway epithelium showed no difference between controls and any of the single or double mutants, suggesting that Dll disruption affected primarily the NEB microenvironment. The frequency of NEBs per airway (%) was largely unaffected, although a small difference in *Dll4* mutants reached statistical significance. However, the number of PNECs per NEB was significantly increased in both the Dll1$^{cnull}$; Dll4$^{cnull}$ and single Dll4$^{cnull}$ airways, indicating that the size of NEBs was dramatically altered in these mutants.

Together the data indicated that the mechanisms that restrict PNEC fate and limit expansion of NEB were severely disrupted in *Dll* mutants.

## Notch signaling and NEB-associated CCs are preserved in the absence of Dll ligands

The strikingly preserved integrity of the NEB microenvironment of double Jag1$^{cnull}$; Jag2$^{cnull}$ mutants led us to hypothesize that Dll1 and Dll4 were not only necessary and sufficient to activate local Notch signaling but also endowed the unique features of the NEB-associated club cells that distinguish them from club cells elsewhere. The absence of Dll ligands in the expanded population of NEB from double *Dll* mutants provided an opportunity to examine this issue. We asked whether the robust activation of Notch signaling seen in NEB-associated club cells (CCs) of control and *Jag* double mutants was also present in Dll1$^{cnull}$; Dll4$^{cnull}$ mice. Double IF for Ascl1 and N1ICD in E18.5 lung sections showed strong N1ICD signals in the NE-associated CCs of mutants indistinguishable from that of controls (*Figure 5A*). Notably, the NEB expansion in Dll1$^{cnull}$; Dll4$^{cnull}$ mutants was accompanied by a respective expansion of the NEB-associated CCs. The identity of these cell populations was further confirmed by expression of Cgrp (NEB) as well as SSEA1 and low *Cyp2f2* (NEB-associated CCs). Double ISH/immunohistochemistry showed the characteristic low levels of *Cyp2f2* expression in NEB-associated cells in contrast to the strong signals outside the NEB microenvironment (*Figure 5B*). The aberrant expansion of the NEB-associated cells was further demonstrated by qPCR analysis of lung homogenates, which showed a significant increase in expression of *Upk3a* in Dll1$^{cnull}$; Dll4$^{cnull}$ mutants compared to controls (*Figure 6A*). Of note, we found no change in expression of markers not directly associated with the NEB microenvironment, such as *Scgb1a1* (CC10) or *Foxj1,* which suggested that the Jag ligands present in double Dll1$^{cnull}$; Dll4$^{cnull}$ lungs were capable of activating Notch and mediating the balance of secretory-ciliated cell differentiation (*Figure 6A*).

Lastly, ISH of E18.5 lungs confirmed the marked expansion in the domain of expression of *Upk3a* in Dll1$^{cnull}$; Dll4$^{cnull}$ mice and their association with NEBs (*Figure 6B–C*). Together the data strongly suggested that, in spite of the inability to express *Dll1* and *Dll4,* key features of the NEB microenvironment are preserved in these mutants by Jag ligand activation of Notch signaling.

## Discussion

Here we provide evidence for distinct roles of Notch ligands once epithelial progenitors initiate differentiation. We show that Jagged-driven Notch signaling differentially regulates cell type-specific programs of cell fate in a temporal and spatial fashion along the developing respiratory tract epithelium. In extrapulmonary airways (largely trachea here) we found that Jag ligands are not required to induce or maintain the fate of basal cell precursors. These precursors are known to generate luminal cells in fetal airways (*Yang et al., 2018*) and we now report that double *Jag1* and *Jag2* deletion leads to a major imbalance between the basal and luminal compartments with an expansion of basal cell precursors. By contrast, loss of both Jag ligands in intrapulmonary airways had no detectable impact on the NEB microenvironment. Unexpectedly, we found that *Dll* inactivation in Dll1$^{cnull}$; Dll4$^{cnull}$ mutants resulted in marked expansion of NEBs and their associated secretory cells.

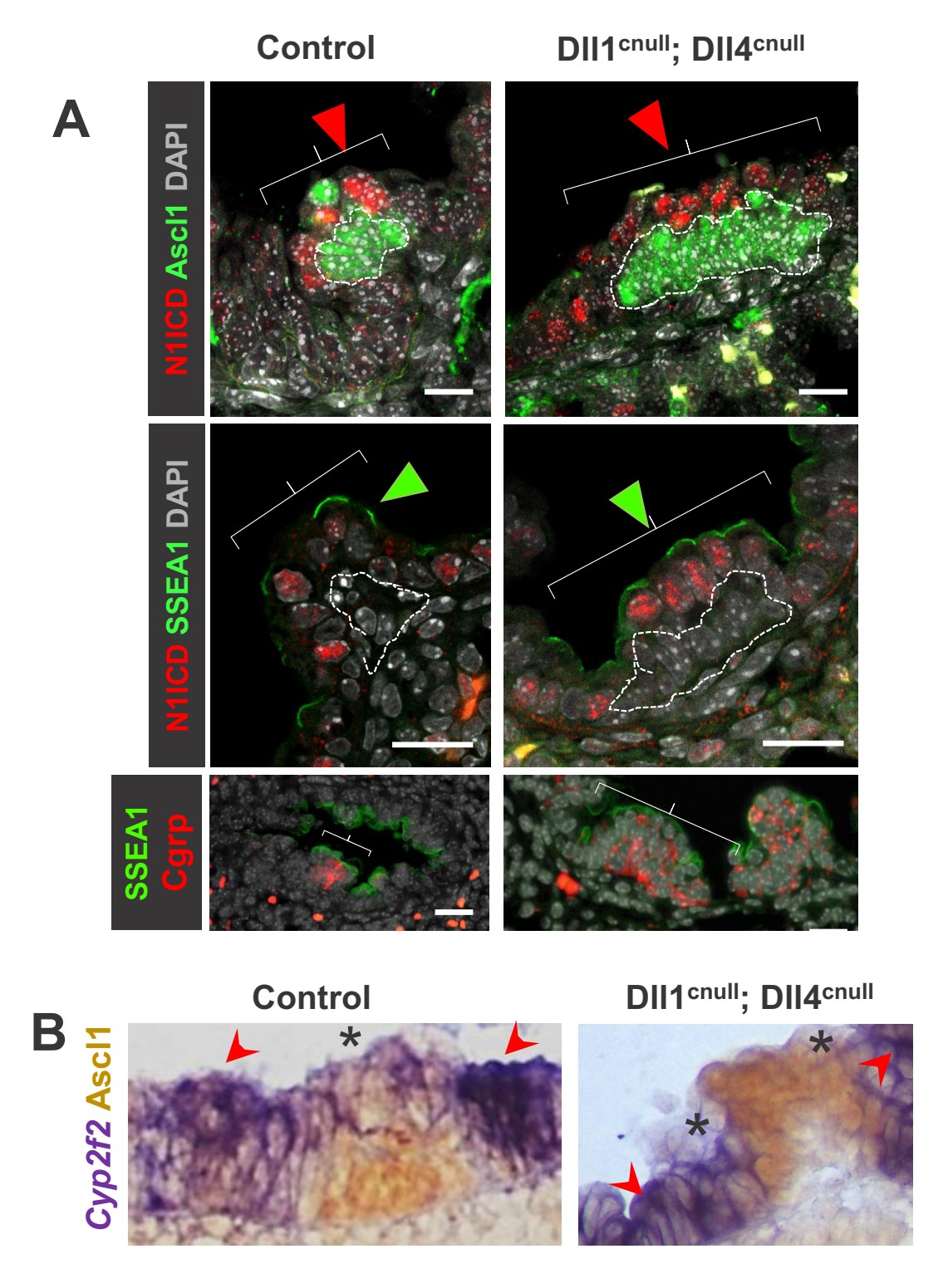

**Figure 5.** Expansion of the NEB-associated Club cells (CC) in the absence Delta ligands. (**A**) Double immunofluorescence of E18.5 control and Dll1cnull; Dll4cnull lungs: Ascl1 and Cgrp expression in control and in the abnormally expanded NEBs (dotted areas) of mutants; robust N1ICD and SSEA1 expression in NEB-associated CCs (arrowheads and brackets). (**B**) Double ISH (*Cyp2f2*)/immunohistochemistry (Ascl1) showing strong *Cyp2f2* expression (arrowheads) in cells flanking the NE microenvironment but only low signals in the NEB-associated CCs (asterisks) of both control and Dll1cnull; Dll4cnull mutants. Note reciprocal high (arrowhead) and low (*) intensity of signals in areas inside and outside the NEB microenvironment. Scale bars in A = 20 μm.

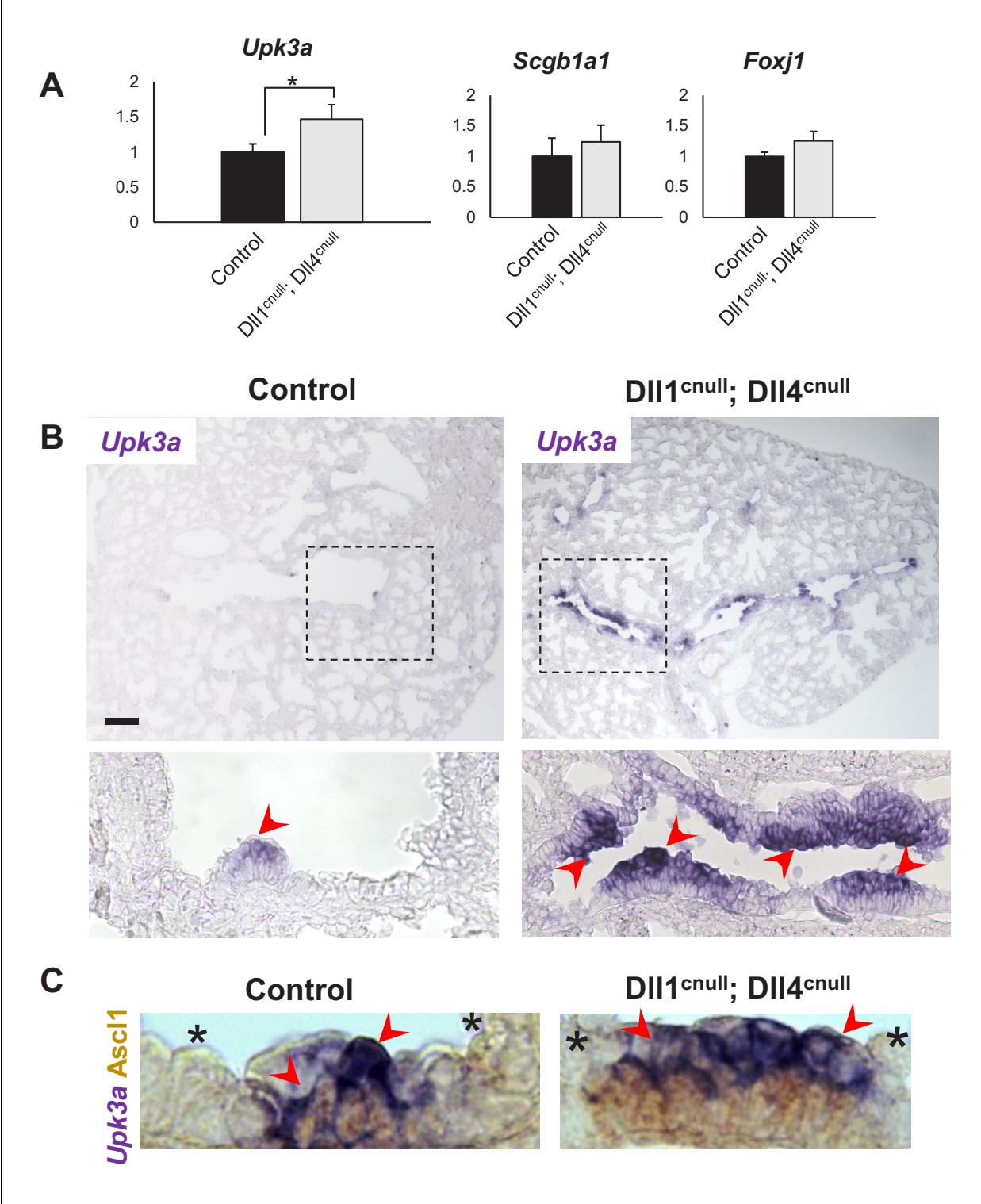

**Figure 6.** Expansion of the *Upk3a* expression domain in Dll1cnull; Dll4cnull lungs. (**A**) qPCR analysis: significantly increased expression of *Upk3a*, but not of *Scgb1a1* or *Foxj1* in mutants relative to controls (n = 3 in both groups). Graphs are mean ± SEM. Student's t-test *p<0.05. (**B**) ISH for *Upk3a* in E18.5 lungs showing marked expansion of the *Upk3a* expression domain (arrowheads) in intrapulmonary airways of Dll1cnull; Dll4cnull mutants (boxed areas enlarged in the lower panels). (**C**) Double immunohistochemistry (Ascl1)/ISH (*Upk3a*) confirms that the *Upk3a+* cells (arrowheads) are NEB-associated CCs. Note reciprocal high (arrowhead) and low (*) intensity of signals in areas outside and inside the NEB microenvironment, respectively. Scale bar in B = 40 µm.

Our analysis of the ontogeny of Notch ligands showed that *Jag2* is expressed well before *Jag1* in the epithelium and that both *Dll1* and *Dll4* appear only after NEBs form in intrapulmonary airways. Establishment of NE vs non-NE fate is known to be associated with induction of Ascl1 and a classic mechanism of lateral inhibition involving activation of Notch-Hes1 in neighboring cells (*Borges et al., 1997*; *Ito et al., 2000*; *Collins et al., 2004*). Interestingly, although Ascl1-labeled NE precursors have been reported in the embryonic lung as early as E12.5, we found no expression of *Dll* (or *Jag*) ligands by these cells prior to E13.5 (*Figure 1*; *Beckers et al., 1999*; *Post et al., 2000*; *Li and Linnoila, 2012*; *Kuo and Krasnow, 2015*). This was intriguing since there is evidence that Hes1 is expressed and already active early in the developing lung epithelium in spite of no evidence of ligand expression in intrapulmonary airways to activate Notch signaling nearby NE cells (*Tsao et al., 2009*; *Noguchi et al., 2015*). This suggests that at these initial stages NE vs. non-NE cell fate selection is mediated by Hes1 in a Notch-independent fashion. Consistent with this, Hes1 deletion in lung epithelial progenitors at the onset of lung development ($Shh^{cre/+}$; $Hes1^{flox/flox}$) results in aberrant expansion of NEB precursors as early as E13.5 (*Noguchi et al., 2015*). Hes1-dependency on Notch is likely established at later developmental stages and could explain why genetic inactivation of the key Notch pathway components Pofut1 or Rbpjk using the ShhCre driver (the same used to delete Hes1, above), had no apparent effect in NE abundance at early (E14.5) stages compared to the severe effects at later (E18.5) stages (*Tsao et al., 2009*) and not shown). Our $Dll1^{cnull}$; $Dll4^{cnull}$ mutants provided the first genetic proof that these ligands are crucially involved in regulating the size the NEBs, with no clear role in controlling other aspects such as the number of PNECs or NEBs per airway. By contrast, Jag ligands had no detectable influence in NEB size or abundance. Even at E14.5, when NEBs are forming along the proximal-distal axis of intrapulmonary airways, *Jag2* is still strongly expressed only at partially overlapping proximal domains such that nascent distal clusters of Ascl1+ cells arise in non-*Jag2* or non-*Jag1*-expressing areas (*Figure 1—figure supplement 2*).

Together our data suggest a model for the role of Notch ligands in lung development (*Figure 7*) in which early, when epithelial progenitors start to differentiate, a wave of Jag-Notch activation initiating in the trachea progresses in a proximal-distal fashion to establish the balance of secretory vs. multiciliated cell fates in intrapulmonary airways. A program of NE cell fate also emerges in intrapulmonary airways and as NEB start to express Dll ligands, Dll-Notch signaling is turned on in adjacent cells to form NEB-associated CCs. Local activation of Notch signaling in these cells shelters the NEB microenvironment from the neighboring epithelium, preventing aberrant NEB expansion. This role is restricted to Dll1 and Dll4, given that *Jag1*, *Jag2* single or double mutants showed no detectable effect in the size of NEBs, NEB-associated CCs or local Notch activation. The NEB microenvironment is maintained by Dll ligands that induce Notch to maintain the local balance of NE and non-NE cell types. The relevance of a late Notch-dependent phase is underscored by the expansion of the NE domain when Dll ligands are unavailable to induce Notch to generate NE-associated CCs locally. Our observations are not in conflict with recent reports that describe aggregation of NEB by a mechanism of NE cell migration (slithering) (*Kuo and Krasnow, 2015*; *Noguchi et al., 2015*; *Branchfield et al., 2016*). Rather we envision that the cell fate specification events described above precede these migratory and cluster-forming events, or overlap at least partially with mechanisms reported here. Intriguingly, in spite of the absence of Dll ligands and having Jag1 and Jag2 as the sole ligands available, the NEB-associated CC in $Dll1^{cnull}$; $Dll4^{cnull}$ airways exhibited robust Notch activation and maintained the unique features of CCs in this microenvironment (strong *Upk3a*, SSEA1, and low *Cyp2f2*). This suggested that the features above do not necessarily depend on the differential activation of Notch by NE-derived Dll ligands and can result from Jag-Notch activation in the cells immediately adjacent to NEBs. We speculate that a currently unidentified cell membrane-associated component of NE cells or a short-range diffusible signal(s) emanating from NEBs modulates Notch signaling or influences the cell fate program in adjacent CCs to endow these features (*Figure 7*).

Lastly, the strength of receptor-ligand interactions is well known to depend on post-translational modifications of Notch receptors, particularly by the family of Fringe proteins, Lunatic (Lfng), Manic (Mfng) and Radical (Rfng) (*Stanley and Okajima, 2010*). Mass spectrometric analysis has demonstrated Lfng to promote Notch activation by Dll1 and decrease its activation by Jag1 (*Kakuda and Haltiwanger, 2017*). Lfng is expressed in NEBs rather than in the NEB-associated CCs, which activates Notch signaling (*Xu et al., 2010b*; *Xu et al., 2010a*). This is reminiscent of the developing intestinal epithelium where Fringe is expressed in the ligand-presenting cells to promote Notch

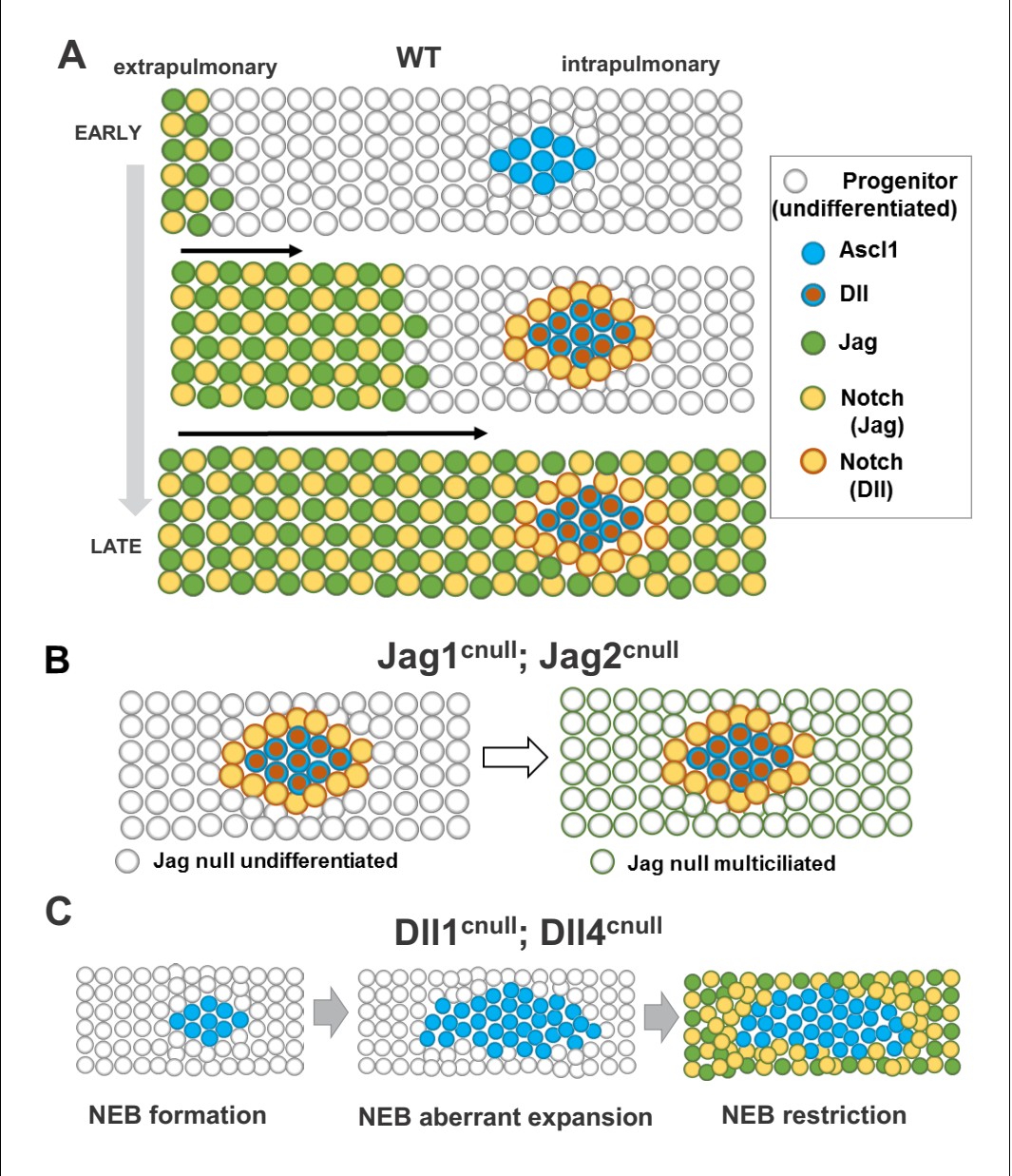

**Figure 7.** Proposed model for ontogeny and function of Notch ligand families in airway epithelial progenitors. In all panels cells expressing Jag (1, 2) or Dll (1, 4) ligands are represented collectively; Notch activated by Jag or Dll are depicted distinctly (box, right). (A) Distinct spatial and temporal onset of expression of Jag and Dll in wild type differentiating airways. *Top row*: Jag expression (green dots) and Jag-Notch activation (yellow dots outlined in green) initiate in trachea/extrapulmonary airways but no Jag or Dll is present in intrapulmonary airways where NE/ NEBs exist (Ascl1, blue circles). *Middle row*: Jag expression continues to advance into intrapulmonary airways in a proximal-to-distal fashion to activate Notch signaling and establish the balance of secretory vs multiciliated cell fates. In intrapulmonary airways Dll ligands are induced in NEBs (red dots outlined in blue) and Dll-Notch signaling is activated in adjacent cells to form NEB-associated club cells (CC, yellow dots outlined in red), limiting NEB expansion. *Bottom row:* As differentiation proceeds, Jag-Notch activation advances more distally and surrounds the NEB and associated CCs. (B) In Jag1[cnull]; Jag2[cnull] mutants the size and composition of the NEB microenvironment in intrapulmonary airways are unaffected since Jag ligands are not required for NE specification or Dll induction. Later, the NEB microenvironment is surrounded by Jag null multiciliated cells and remains unperturbed (right). (C) Dll1[cnull]; Dll4[cnull] double mutants: NE/NEB formation is initiated in intrapulmonary airways (left, compare to A). However the inability of Ascl1 cells to induce Dll ligands prevents induction of Notch-Dll signaling in surrounding undifferentiated progenitors, leading to aberrant expansion of NEBs (middle). Later, as
*Figure 7 continued on next page*

*Figure 7 continued*

Jag expression advances in intrapulmonary airways, Jag-Notch is activated in the progenitor cells surrounding the NEB, generating CCs and restricting its expansion (right, see Discussion).

activity in the neighboring cells (*Kadur Lakshminarasimha Murthy et al., 2018*). There is currently no evidence that these proteins influence epithelial Notch signaling in the developing lung epithelium (*van Tuyl et al., 2005*; *Xu et al., 2010b*; *Xu et al., 2010a*).

In summary our study provides novel insights into developmental mechanisms mediated by Jag/Dll/Notch in the lung. These observations could be of significance in studies of human conditions associated with aberrant expansion or differentiation of NEBs and their associated CCs. Indeed, analysis of human biopsies from normal donors and patients with pulmonary NE cell hyperplasias suggest that both the NE and NE-associated CC components are coordinately altered (*Guha et al., 2017*). Further studies examining the impact of Notch ligands and downstream signals in these diseases are likely to provide important insights into their pathogenesis.

# Materials and methods

**Key resources table**

| Reagent type (species) or resource | Designation | Source or reference | Identifiers | Additional information |
|---|---|---|---|---|
| Genetic reagent (*M. musculus*) | *Dll1^flox* | PMID: 16495313 | MGI: 3044907 | Dr. Julian Lewis |
| Genetic reagent (*M. musculus*) | *Dll4^flox* | PMID: 18824585 | MGI: 3828266 | Dr. Freddy Radtke (École polytechnique fédérale de Lausanne) |
| Genetic reagent (*M. musculus*) | *Jag1^flox* | PMID: 16495313 | MGI: 3623344 | Dr. Julian Lewis |
| Genetic reagent (*M. musculus*) | *Jag2^flox* | PMID: 20533406 | MGI: 4829504 | Dr. Thomas Gridley (Tufts University) |
| Genetic reagent (*M. musculus*) | *Shh^cre* | Jackson Laboratory | Stock# 005622, RRID: IMSR_JAX:005622 | PMID: 15315763 |
| Antibody | anti-beta IV tubulin (mouse monoclonal) | Abcam | Cat#ab11315, RRID:AB_297919 | IF (1:100) |
| Antibody | anti-Ascl1 (mouse monoclonal) | Thermo Fisher Scientific | Cat# 14-5794-82, RRID: AB_2572887 | IHC/IF (1:100), IF requires tyramide amplification |
| Antibody | anti-CC10 (goat polyclonal) | Santa Cruz Biotechnology | Cat# sc-9772, RRID: AB_2238819 | IF (1:150) |
| Antibody | anti-Cgrp (rabbit polyclonal) | Sigma-Aldrich | Cat# C8198, RRID: AB_259091 | IF (1:2500) |
| Antibody | anti-Foxj1 (mouse monoclonal) | Thermo Fisher Scientific | Cat# 14-9965-82, RRID: AB_1548836 | IHC/IF (1:50) |
| Antibody | anti-Ki67 (rabbit monoclonal) | Cell Signaling Technology | Cat # 9129, RRID: AB_2687446 | IF (1:100) |

*Continued on next page*

*Continued*

| Reagent type (species) or resource | Designation | Source or reference | Identifiers | Additional information |
|---|---|---|---|---|
| Antibody | anti-Krt5 (rabbit polyclonal) | Biolegend | Cat# 905501, RRID: AB_2565050 | IF (1:500) |
| Antibody | anti-Krt8 (chicken polyclonal) | Abcam | Cat# ab107115, RRID: AB_10976462 | IF (1:500) |
| Antibody | anti-N1ICD (rabbit monoclonal) | Cell Signaling Technology | Cat# 4147, RRID: AB_2153348 | IF (1:100), requires tyramide amplification |
| Antibody | anti-p63 (rabbit polyclonal) | Santa Cruz Biotechnology | Cat# sc8343, RRID: AB_653763 | IHC/IF (1:400) |
| Antibody | anti-Scgb3a2 (goat polyclonal) | R and D Systems | Cat# AF3465, RRID: AB_2183550 | IF (1:100) |
| Antibody | anti-SSEA1 (mouse monoclonal) | EMD Millipore | Cat# MAB4301, RRID: AB_177627 | IF (1:300) |
| Antibody | anti-rabbit, Alexa Fluor 488 (donkey polyclonal) | Thermo Fisher Scientific | Cat# A21206, RRID: AB_2535792 | IF (1:300) |
| Antibody | anti-rabbit, Alexa Fluor 568 (donkey polyclonal) | Thermo Fisher Scientific | Cat# A10042, RRID: AB_2534017 | IF (1:300) |
| Antibody | anti-rabbit, Alexa Fluor 647 (donkey polyclonal) | Thermo Fisher Scientific | Cat# A31573, RRID: AB_2536183 | IF (1:300) |
| Antibody | anti-goat, Alexa Fluor 488 (donkey polyclonal) | Thermo Fisher Scientific | Cat# A11055, RRID: AB_2534102 | IF (1:300) |
| Antibody | anti-goat, Alexa Fluor 568 (donkey polyclonal) | Thermo Fisher Scientific | Cat# A11057, RRID: AB_2534104 | IF (1:300) |
| Antibody | anti-goat, Alexa Fluor 647 (donkey polyclonal) | Thermo Fisher Scientific | Cat# A21447, RRID: AB_2535864 | IF (1:300) |
| Antibody | anti-chicken, Alexa Fluor 488 (donkey polyclonal) | Jackson ImmunoResearch | Cat# 703-545-155, RRID: AB_2340375 | IF (1:300) |
| Chemical compound, drug | BM-Purple | Roche | Cat# 11442074001 | |
| Commercial assay, kit | anti-Mouse IgG (Peroxidase) polymer detection kit, made in horse | Vector Laboratories | Cat# MP-7402, RRID: AB_2336528 | |
| Commercial assay, kit | anti-Rabbit IgG (Peroxidase) polymer detection kit, made in horse | Vector Laboratories | Cat# MP-7401, RRID: AB_2336529 | |
| Commercial assay, kit | TSA Plus Cyanine 3 | Akoya Biosciences | Cat# NEL753001KT | Previously PerkinElmer |
| Commercial assay, kit | TSA Plus Cyanine 5 | Akoya Biosciences | Cat# NEL745001K | Previously PerkinElmer |

*Continued on next page*

*Continued*

| Reagent type (species) or resource | Designation | Source or reference | Identifiers | Additional information |
|---|---|---|---|---|
| Commercial assay, kit | ImmPACT DAB peroxidase (HRP) substrate | Vector Laboratories | Cat# SK-4105, RRID: AB_2336520 | |
| Commercial assay, kit | RNeasy mini kit | Qiagen | Cat# 74104 | |
| Commercial assay, kit | MAXIscript T7 | Thermo Fisher Scientific | Cat# AM1314M | |
| Commercial assay, kit | SuperScript IV First-Strand Synthesis System | Thermo Fisher Scientific | Cat# 18091050 | |

## Mouse models

$Dll1^{flox/flox}$ and $Jag1^{flox/flox}$ mice were provided by Dr. Julian Lewis (*Hozumi et al., 2004*; *Brooker et al., 2006*). Dll1$^{cnull}$ mice were generated by crossing $Dll1^{flox/flox}$ female mice with $Dll1^{flox/+}$; $Shh^{cre/+}$ males. $Dll4^{flox/flox}$ mice were obtained from Dr. Freddy Radtke (*Koch et al., 2008*). Dll4$^{cnull}$ mice were generated by crossing $Dll4^{flox/flox}$ female mice with $Dll4^{flox/+}$; $Shh^{cre/+}$ males. Dll1$^{cnull}$; Dll4$^{cnull}$ mice were generated by crossing $Dll1^{flox/flox}$; $Dll4^{flox/flox}$ females with $Dll1^{flox/+}$; $Dll4^{flox/+}$; $Shh^{cre/+}$ males. Jag1$^{cnull}$ mice were generated by crossing $Jag1^{flox/flox}$ females with $Jag1^{flox/+}$; $Shh^{cre/+}$ males. $Jag2^{flox/flox}$ mice were provided by Dr. Thomas Gridley (*Xu et al., 2010b*). Jag2$^{cnull}$ mice were generated by crossing $Jag2^{flox/flox}$ females with $Jag2^{flox/+}$; $Shh^{Cre/+}$ males. Jag1$^{cnull}$; Jag2$^{cnull}$ mice were generated by crossing $Jag1^{flox/flox}$; $Jag2^{flox/flox}$ females with $Jag1^{flox/+}$; $Jag2^{flox/+}$; $Shh^{cre/+}$ males. Embryos were harvested at E14.5 and E18.5, where day 0.5 was counted as the morning when a vaginal plug was found. All experiments involving animals were performed in accordance with the protocols approved by Columbia University Medical Center.

## Immunofluorescence

Whole lung and trachea were harvested from mice at E14.5 and E18.5 and fixed in 4% paraformaldehyde at 4°C overnight. Samples then underwent PBS washes and 15% and 30% sucrose washes before embedding in OCT. Samples were incubated with primary antibodies (overnight at 4°C) and secondary antibodies conjugated with Alexa488, 568, or 647 (1:300) with NucBlue Fixed Cell ReadyProbes Reagent (DAPI) (Thermo Fisher #R37606) for 45 min. After washing, samples were mounted with ProLong Gold antifade reagent for analysis. When necessary, heat-induced epitope retrieval was performed using citric acid-based antigen unmasking solution (Vector Laboratories #H-3300). Ascl1 and N1ICD staining required tyramide amplification (cyanine 3 or cyanine 5) used with horse radish peroxidase conjugation (species-specific ImmPRESS kit, Vector Laboratories). Antibodies used were: anti-β-tubulin IV (Abcam #ab11315, 1:100), anti-Ascl1 (Thermo Fisher #14-5794-82, 1:100), anti-CC10 (Santa Cruz sc9772, 1:150), anti-Cgrp (Sigma Aldrich #C8198, 1:2500), anti-Foxj1 (Thermo Fisher #14-9965-80, 1:50), anti-Ki67 (Cell Signaling #9129, 1:100), anti-Krt5 (Biolegend #905501, 1:500), anti-Krt8 (Abcam #ab107115, 1:500); anti-N1ICD (Cell Signaling #4147, 1:100), anti-p63 (Santa Cruz #sc8343, 1:400), anti-Scgb3a2 (R and D Systems #AF3465, 1:100), anti-SSEA1 (EMD Millipore #MAB4301, 1:300). Images were acquired using a Leica DMi8 microscope or Zeiss LSM710 confocal laser scanning microscope.

## Morphometric analysis

To determine the percentage of Foxj1+ ciliated cells in control and *Jag*-cnull mutant intrapulmonary airways E18.5 coronal sections of whole lungs were stained with Foxj1 and DAPI. Two sections from two separate embryos for each genotype were used for counting. DAPI+ epithelial cells were counted in intrapulmonary airways and were compared to the number of Foxj1+ cells to determine Foxj1+ percentages.

To determine the percentage of Foxj1+ ciliated cells in control and *Jag*-cnull mutant tracheas E18.5 coronal sections of whole tracheas were stained with Foxj1 and DAPI. Two sections of whole trachea from one embryo for each genotype were used for counting. DAPI+ epithelial cells were counted in one side of the trachea and were compared to the number of Foxj1+ cells to determine Foxj1+ percentages.

Analysis of neuroendocrine cells and neuroepithelial bodies (NEBs) was performed on E14.5 *Delta*-cnull mutants. Sections were stained with Ascl1 and DAPI. Three sections from three separate embryos for each genotype were used for counting. For each section the number of intrapulmonary airways was counted, as well as the number of NEBs and solitary neuroendocrine cells. The ratios of NEBs/airway and solitary neuroendocrine cells/airway were calculated. Additionally, NEB size was examined. In each section, NEB size was determined by counting the number of neuroendocrine cells in contact with each other, where an NEB was determined to be a group of three or more cells.

## In situ hybridization

Frozen sections were processed as described for immunofluorescence. In situ hybridization was performed using digoxigenin-UTP-labeled probes as previously described (*Tsao et al., 2008*; *Tsao et al., 2009*; *Guha et al., 2012*). Probes are listed in *Table 1*. Hybridization probes were ordered from Integrated DNA Technologies at 25 nM with standard desalting and stored as 100 μM stocks in DEPC-treated water.

## Quantitative real-time PCR

Quantitate real-time PCR was performed as previously described (*Tsao et al., 2009*). RNA was extracted using the RNeasy kit (Qiagen) and reverse transcribed using Oligo(DT) primers (Super-Script III or IV kits, Thermo Fisher). The following primers (Thermo Fisher) were used: Ascl1 (Mm03058063_m1), Cgrp (Mm00801463_g1), Foxj1 (Mm01267279_m1), Scgb1a1/CC10 (Mm00442046_m1), Scgb3a2 (Mm00504412_m1), Upk3a (Mm00452321_m1). Reactions were performed using Taq-Man Advanced Master Mix (Thermo Fisher #4444556) using β-actin as internal control and a Step-One Plus Instrument (Applied Biosystems). ΔΔCT method was used to calculate changes in expression levels.

**Table 1.** Primers used to generate ISH probes.

T7 or T3 primers were added to the reverse or forward primers of each gene, respectively.

| Gene | Forward (5' → 3') | Reverse (5' → 3') |
|---|---|---|
| *Dll1* | AATTAACCCTCACTAAAGGG AGACTGCTGA GAGAGGAAGGGAG | TAATACGACTCACTA TAGGGAGAAGACCCGAA GTGCCTTTGTA |
| *Dll4* | AATTAACCCTCAC TAAAGGGAGACTACT CAGACACCCAGCTCC | TAATACGACTCACT ATAGGGAGAATCCTTTGC AAGCCTCCTCT |
| *Jag1* | AATTAACCCTCACTAAAG GGAGACGCCATAG GTAGAGTTTGAGG | TAATACGACTCACTA TAGGGAGATAGTTGCTGT GGTTCTGAGC |
| *Jag2* | AATTAACCCTCACTAA AGGGAGATGGCACC CAGAACCCTTG | TAATACGACTCA CTATAGGGAGAATACT CCGTTGTTTTCCGCC |
| *Ascl1* | AATTAACCCTCACT AAAGGGTCGTCCTCTC CGGAACTGAT | TAATACGACTCACTAT AGGGAGAAGAAGCAAA GACCGTGGGAG |
| *Upk3a* | AATTAACCCTCAC TAAAGGGGTGGCT GGACTGTGAACCTC | TAATACGACTCA CTATAGGGTTGCCCA CCCTGACTAGGTA |
| *Cyp2f2* | AATTAACCCTCA CTAAAGGGGGAACTT TGGAGGCATGAAA | TAATACGACTCAC TATAGGGAACTCCTGAG GCGTCTTGAA |

## Acknowledgements

We thank Cathy Mendelsohn, Michael Shen, Mitsuru Morimoto and members of the Cardoso lab for valuable scientific discussions. We are indebted to Arjun Guha, who contributed for the initiation of this project. We also thank Jun Qian, Anne Hynes and Jingshu Huang for technical assistance.

## Additional information

### Funding

| Funder | Grant reference number | Author |
| --- | --- | --- |
| National Heart, Lung, and Blood Institute | R35-HL135834-01 | Wellington V Cardoso |

The funders had no role in study design, data collection and interpretation, or the decision to submit the work for publication.

### Author contributions

Maria R Stupnikov, Conceptualization, Data curation, Formal analysis, Validation, Investigation, Visualization, Methodology, Writing—original draft, Writing—review and editing; Ying Yang, Munemasa Mori, Data curation, Formal analysis; Jining Lu, Conceptualization, Writing—review and editing, Contributed to analysis of mouse phenotypes and interpretation of results; Wellington V Cardoso, Conceptualization, Resources, Formal analysis, Supervision, Funding acquisition, Investigation, Project administration, Writing—review and editing

### Author ORCIDs

Maria R Stupnikov (iD) https://orcid.org/0000-0002-5349-5233
Wellington V Cardoso (iD) https://orcid.org/0000-0002-8868-9716

### Ethics

Animal experimentation: This study was performed in strict accordance with the recommendations in the Guide for the Care and Use of Laboratory Animals of the National Institutes of Health. All of the animals were handled according to approved institutional animal care and use committee (IACUC) protocol (AAAS0503) of Columbia University.

### Decision letter and Author response

Decision letter https://doi.org/10.7554/eLife.50487.sa1
Author response https://doi.org/10.7554/eLife.50487.sa2

## Additional files

### Supplementary files

• Transparent reporting form

### Data availability

All data generated or analyzed have been included in the manuscript and supporting files. No databases have been generated in this study.

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
