## [Decision Letter]

Thank you for submitting your article "Jagged and Delta ligands control distinct events during airway progenitor cell differentiation" for consideration by *eLife*. Your article has been reviewed by three peer reviewers, one of whom is a member of our Board of Reviewing Editors, and the evaluation has been overseen by a Reviewing Editor and Didier Stainier as the Senior Editor. The reviewers have opted to remain anonymous.

The reviewers have discussed the reviews with one another and the Reviewing Editor has drafted this decision to help you prepare a revised submission.

Summary:

All of the reviewers found your study to be insightful and important. Reviewer 1 had concerns regarding some of the interpretations of the role of the Jagged ligands which may require some additional experimentation. Moreover, reviewers 2 and 3 had concerns on the timing and quantitative nature of the roles for Jagged and Delta ligands which may also require additional experiments to address. The additional experimentation should be well within the authors abilities. Please also revise the text to address the noted grammatical issues.

*Reviewer #1:*

This is an interesting report from a group that has been at the forefront of exploring the role of Notch signaling in airway development. The authors have provided evidence of a divergence in the role of specific Notch ligands with Jag ligands regulating the balance of luminal cell development whereas delta ligands regulating neuroendocrine cell development. There are a few things that need to be done to strengthen the report:

1) The data in Figure 3 showing that loss of Jag1/Jag2 is a bit confusing at least how it is described. The authors state that there is little to no effect of these 2 ligands on NEB formation. Yet their data show a strong decrease in *Upk3a* expression and their ISH shows what appears to be a reduction in the number of NEBs. The authors should clarify whether loss of Jag1/Jag2 lead to a quantitative loss of NEBs and what this means in the context of differential Notch ligand usage in lung development.

2) The data in Figure 4 and Figure 5 showing expansion of NEBs upon loss of Dll1/Dll4 are a bit confusing. Do the authors think this is really an expansion of NEBs or could it be a reprogramming of airway epithelium into a neuroendocrine fate rather than secretory or ciliated? It is difficult to tell in the pictures provided. This is an important comparison as it could lead to an even more interesting result.

3) The *Upk3a* staining in Figure 5C is not very convincing. It needs to be improved upon or another technique used (i.e. RNAscope) to really define the differences between WT and Dll1/Dll4 knockouts.

4) There are numerous grammatical errors throughout the manuscript that should be corrected.

*Reviewer #2:*

This is an interesting article that explores roles for various Notch ligands in the regulation of epithelial cell fate in the developing mouse lung. These studies build upon a fairly extensive literature published by these and other authors. This is important work but even though the presented data are of good quality, conclusions seem overstated and data raise unexplained questions that should be addressed with additional experimental analysis and/or discussion. The following concerns were noted:

1) Figure 1. Data presented in Figure 1 show in situ localization of Jag1, Jag2, Dll1 and Dll4 mRNA within embryonic mouse lungs at either E12, 12.5 or E13.5 (Jag1 and Jag 2) or E13.5 and E14.5. Even though these data provide information regarding spatial distribution of these ligands, it does not adequately address differences in the temporal expression pattern that the authors suggest represents an important distinction between these Notch ligands. As such, conclusions such as those made in multiple statements such as "Analyses of the ontogeny of these ligands showed that Jag2 is expressed well before all others and that Dll ligands appear last.…." are not supported by data presented in this manuscript. If temporal/spatial expression data are fragmented and spread across a number of other manuscripts in the existing literature, it would be helpful to cite these papers and perhaps present a comprehensive analysis in this manuscript.

2) Results from Figure 1 and Figure 2 present a rather confusing picture that most likely results from the fact that experiments in Figure 1 only evaluate Jag ligand expression from E12-E14.5 yet results of *Shh*-cre mediated loss-of-function are evaluated at E18.5. A more comprehensive presentation of spatial/temporal expression in Figure 1 would allow readers to better appreciate why Jag2-lof has minimal impact on lung development compared to Jag1-lof (which according to data in Figure 1 is only expressed within developing lung vasculature and not endoderm)?

3) Related to concerns above, the authors conclude that "We propose that Jag1-Notch mediated signaling compensates for the loss of Jag2, preserving most of the secretory cell population in these mutants". However, no data are shown regarding Jag1 expression in either later stages of normal developing lung or following Jag2-lof.

4) Figure 1—figure supplement 2 – title states that NEB's appear in "non-Jag2 expressing" domains. In fact, the data suggest that appearance of NEB's occurs independently of Jag2 expression as neither presence nor absence of Jag2 mRNA impacts presence of clusters of Ascl1-expressing cells.

5) The Discussion section is written in a rather cryptic style that is difficult to read. For example, the statement "While our findings are not in conflict with the reported role of Hes1-Ascl1 in NE-non-NE cell fate choice, they strongly suggest that this mechanism occurs in a Notch-independent fashion." is not adequately discussed. Please reference the article being referred to and also explain why their data strongly suggest that this mechanism occurs in a Notch-independent fashion. Perhaps related to this, an interesting but unresolved observation in the current study is that it appears that even though Dll ligands activate Notch signaling in variant club cells that limit expansion of NEB's, this occurs despite the persistence of N1ICD immunoreactivity within this population. What about other Notch receptors? Is it possible that N1ICD results from Jag signaling? Further discussion and possibly experimental analysis would be of value in clarifying these issues for the reader.

6) The statement "These mechanisms likely contribute to the pathogenesis of human conditions resulting from aberrant lung differentiation, including NE hyperplasias and cancer" are not supported by data and seem rather speculative.

*Reviewer #3:*

This paper determines the specific roles of Notch ligand family of genes in airway epithelial fate determination during lung development. The authors showed that Dll and Jag start to express in different temporal and spatial manner. While Jag1 and Jag2 are crucial for multiciliated cell differentiation in the lung and basal cell and multiciliated cell differentiation in the trachea, loss of Dll1 and Dll4 led to expansion of PNECs. The authors also assayed the molecular characteristics of the NEB-associated club cells in both Dll^cnull^ and Jag^cnull^ mutants, providing a novel insight into their initiation. Overall, this manuscript presents a systemic characterization of the distinct functions of Jag/Dll/Notch in lung epithelium, and should raise broad interest.

Specific points:

- What may be the reason that loss of Jag1, but not Jag2, led to significant multiciliated cell phenotype, given the fact that Jag2 is the predominant Notch ligand in the extrapulmonary airway by in situ? qRT would be informative of the level.

- What may be the reason behind the role of Jag1 in ciliated cells, and Jag2 in basal cells? Temporal expression of Jag1, Jag2 at E15.5-16.5 would be informative.

- A close analysis of Jag expression in the NEB microenvironment in the Dll mutant will be informative.

- Subsection “DLL ligands control the size and of the NEB microenvironment”: "Interestingly, Ki67 staining.…… in proliferation" is based on data at E18.5. Ki67 staining at earlier stage (e.g. E14.5) would further support the conclusion.

- Is there any alveolar phenotype in any of the mutants?

---

## [Author Response]

Reviewer #1:1) The data in Figure 3 showing that loss of Jag1/Jag2 is a bit confusing at least how it is described. The authors state that there is little to no effect of these 2 ligands on NEB formation. Yet their data show a strong decrease in Upk3a expression and their ISH shows what appears to be a reduction in the number of NEBs. The authors should clarify whether loss of Jag1/Jag2 lead to a quantitative loss of NEBs and what this means in the context of differential Notch ligand usage in lung development.

Indeed, we have not emphasized in our text that the *Upk3a* expression depicted in Figure 3E by qPCR analysis in whole lungs included *Upk3a* expression from NEB-associated CCs and the multiple small foci of *Upk3a* scattered in airways. Since Ascl1 qPCR analysis in the same samples and IF for Cgrp confirmed that NEB population was equally preserved in control and double Jag mutants (Figure 3A), the significant decrease in *Upk3a* was unlikely due to the loss of the NEB-associated CCs but rather resulted from the loss of the *Upk3a*-expressing scattered cell population. This was further supported by the evidence of preserved NEB-associated SSEA1 population (Figure 3B-D) and the fact that *Upk3a* requires Notch activation not present in the extensive areas devoid of Jag ligands. These explanations have been added in the current revised version.

2) The data in Figure 4 and Figure 5 showing expansion of NEBs upon loss of Dll1/Dll4 are a bit confusing. Do the authors think this is really an expansion of NEBs or could it be a reprogramming of airway epithelium into a neuroendocrine fate rather than secretory or ciliated? It is difficult to tell in the pictures provided. This is an important comparison as it could lead to an even more interesting result.

Given the expanded domains of expression of NEB and NEB-associated markers without evidence of increased proliferation, we ascribed these changes to the activation of an aberrant differentiation program in which NEB expansion was no longer restricted by Dll induction of Notch signaling in adjacent cells. This restriction occurred only later, when the expanded NEB was surrounded by Jag-Notch activation (independent of Dll). We do not believe that this results from reprograming (switch in cell fate) but rather the induction of an aberrant program of cell fate selection. We have commented on these issues in the revised Discussion section.

3) The Upk3a staining in Figure 5C is not very convincing. It needs to be improved upon or another technique used (i.e. RNAscope) to really define the differences between WT and Dll1/Dll4 knockouts.

We now provide new additional evidence of major expansion in the *Upk3a* domain of Dll1^cnull^; Dll4^cnull^ mutants (new Figure 6B).

4) There are numerous grammatical errors throughout the manuscript that should be corrected.

We have edited the text extensively to address this issue.

Reviewer #2:1) Figure 1. Data presented in Figure 1 show in situ localization of Jag1, Jag2, Dll1 and Dll4 mRNA within embryonic mouse lungs at either E12, 12.5 or E13.5 (Jag1 and Jag 2) or E13.5 and E14.5. Even though these data provide information regarding spatial distribution of these ligands, it does not adequately address differences in the temporal expression pattern that the authors suggest represents an important distinction between these Notch ligands. As such, conclusions such as those made in multiple statements such as "Analyses of the ontogeny of these ligands showed that Jag2 is expressed well before all others and that Dll ligands appear last.…." are not supported by data presented in this manuscript. If temporal/spatial expression data are fragmented and spread across a number of other manuscripts in the existing literature, it would be helpful to cite these papers and perhaps present a comprehensive analysis in this manuscript.

As pointed out by the reviewer, other papers, including some from our lab, have already reported the Jag and Dll ligand expression patterns in the developing lung at late developmental stages. We focused on the relatively less explored early stages of lung development and initially did not include stages already reported. However, we agree that, as presented, the information seemed fragmented and difficult to interpret. Thus, in our revised manuscript we now provide a more comprehensive expression pattern analysis with stages ranging from E12-E18.5. We also include a diagram that summarizes the timing and spatial distribution in the respiratory epithelium (revised Figure 1 and Figure 1—figure supplement 1).

2) Results from Figure 1 and Figure 2 present a rather confusing picture that most likely results from the fact that experiments in Figure 1 only evaluate Jag ligand expression from E12-E14.5 yet results of Shh-cre mediated loss-of-function are evaluated at E18.5. A more comprehensive presentation of spatial/temporal expression in Figure 1 would allow readers to better appreciate why Jag2-lof has minimal impact on lung development compared to Jag1-lof (which according to data in Figure 1 is only expressed within developing lung vasculature and not endoderm)?

As described above, in our revised manuscript we now provide a comprehensive expression pattern analysis (E12-E18.5) of ligands and a summary diagram. Indeed, these show more clearly that the Jag1 is the predominant ligand in intrapulmonary airways, explaining the differences in phenotype between the Jag mutants.

3) Related to concerns above, the authors conclude that "We propose that Jag1-Notch mediated signaling compensates for the loss of Jag2, preserving most of the secretory cell population in these mutants". However, no data are shown regarding Jag1 expression in either later stages of normal developing lung or following Jag2-lof.

To address this criticism we now include Jag ligand expression analysis at later stages (revised Figure 1—figure supplement 1) and show that Jag1 expression is extensively expressed in intrapulmonary airways, only partially overlapping with Jag2. We also show expression of Jag1 in the airway epithelium of Jag2^cnull^ mice (new Figure 2). These support the idea that Jag1 maintains Notch signaling in intrapulmonary airways as an attempt to preserve the secretory cell population in Jag2 mutants. We have commented on these observations in the revised text.

4) Figure 1—figure supplement 2 – title states that NEB's appear in "non-Jag2 expressing" domains. In fact, the data suggest that appearance of NEB's occurs independently of Jag2 expression as neither presence nor absence of Jag2 mRNA impacts presence of clusters of Ascl1-expressing cells.

The reviewer is correct. However, we included this picture rather to illustrate the discrepancy between sites of Jag2 and Ascl1 expression along the proximal-distal axis of intrapulmonary airway as NEBs form more distally.

5) The Discussion section is written in a rather cryptic style that is difficult to read. For example, the statement "While our findings are not in conflict with the reported role of Hes1-Ascl1 in NE-non-NE cell fate choice, they strongly suggest that this mechanism occurs in a Notch-independent fashion." is not adequately discussed. Please reference the article being referred to and also explain why their data strongly suggest that this mechanism occurs in a Notch-independent fashion.

We have revised the text extensively and discuss in more detail the Notch-independent mechanism proposed.

Perhaps related to this, an interesting but unresolved observation in the current study is that it appears that even though Dll ligands activate Notch signaling in variant club cells that limit expansion of NEB's, this occurs despite the persistence of N1ICD immunoreactivity within this population. What about other Notch receptors? Is it possible that N1ICD results from Jag signaling? Further discussion and possibly experimental analysis would be of value in clarifying these issues for the reader.

We would like to stress that the inability of Dll mutants to limit NEB expansion is an early event that results from the failure to activate Dll-mediated Notch signaling locally. Indeed, in Dll1^cnull^; Dll4^cnull^ mutants the expanded NEBs are later surrounded by club cells that express NICD as a result of a proximal-distal “wave” of Jag-activation of Notch signaling for secretory-ciliated cell fate selection in airway progenitor cells. These ideas and model are presented in Figure 7 (“Later, Jag ligands in intrapulmonary airways activate Notch signaling in progenitor cells surrounding the NEB and generate typical NEB-associated CCs”). Other Notch receptors besides N1ICD are likely involved in these events as shown by genetic studies in which Notch1-3 were sequentially inactivated in the developing lung (Morimoto et al., 2012). We did not explore further the role of Notch receptors as we considered it be beyond of the scope of the present work.

6) The statement "These mechanisms likely contribute to the pathogenesis of human conditions resulting from aberrant lung differentiation, including NE hyperplasias and cancer" are not supported by data and seem rather speculative.

We agree and have edited the abstract and discussion to address this issue.

Reviewer #3:- What may be the reason that loss of Jag1, but not Jag2, led to significant multiciliated cell phenotype, given the fact that Jag2 is the predominant Notch ligand in the extrapulmonary airway by in situ? qRT would be informative of the level.

Our data is consistent with the reviewer’s idea that this is due to the predominant expression of Jag1 in intrapulmonary airways mediating secretory vs ciliated cell fate choice. Thus, Jag1 mutants have unbalanced excessive abundance of ciliated cells at the costs of secretory cells in intrapulmonary airways. The differences in Jag patterns of expression are now more clearly depicted in current Figure 1—figure supplement 1.

- What may be the reason behind the role of Jag1 in ciliated cells, and Jag2 in basal cells? Temporal expression of Jag1, Jag2 at E15.5-16.5 would be informative.

Similarly, as above, this could be attributed to differences in the spatial distribution of the Jag ligands. Jag 1 is essentially expressed in intrapulmonary airways, which do not contain basal cells. By contrast Jag2 is prominent in the tracheal epithelium from the earliest stages, where basal cell precursors undergo differentiation. We have included Jag2 expression in E15.5-E16.5, as requested, and show Jag2 co-localization with the basal cell marker p63 (Figure 1—figure supplement 1C; Jag1 is not expressed in trachea).

- A close analysis of Jag expression in the NEB microenvironment in the Dll mutant will be informative.

Previous studies from our lab and from our colleagues, as well as data from our current analysis showed Jag ligand expression, mostly Jag1, throughout the epithelium of intrapulmonary airways, including the regions immediately flanking the NEB microenvironment (Tsao et al., 2009; Xu et al., 2009; Figure 1—figure supplement 1, Figure 1—figure supplement 2). Moreover, we had evidence that Dll1^cnull^; Dll4^cnull^ mutants have a normal balance of secretory-ciliated cells in intrapulmonary airways, as seen by analysis of markers of differentiation that are largely dependent on normal Jag-Notch signaling (Figure 6A: Scgb1a1, Foxj1). These observations leave Jag as the only ligands acting in Dll1^cnull^; Dll4^cnull^ mutants to mediate Notch-related events nearby and away from the NEB microenvironment.

- Subsection “DLL ligands control the size and of the NEB microenvironment”: "Interestingly, Ki67 staining.…… in proliferation" is based on data at E18.5. Ki67 staining at earlier stage (e.g. E14.5) would further support the conclusion.

As requested, we have performed Ki67 staining in double E14.5 Dll null mutants and confirmed our previous conclusion that cell proliferation is not increased in the areas of expanded NEBs (current Figure 4C).

- Is there any alveolar phenotype in any of the mutants?

We have limited our study to prenatal stages (prior to alveolar development) largely due to neonatal lethality. Our analysis of E18.5 mutants did not detect obvious changes in sacculation or distal differentiation.